# Deep-ComAIR: A Framework for Predicting TCR-pMHC Binding through Complex Structural Analysis

## Abstract

The binding process between T cell receptor (TCR) and peptide-major histocompatibility complex (pMHC) is a fundamental mechanism in adaptive immunity. Current research on binding prediction primarily emphasizes the sequence and structural features of critical regions within these molecules, often neglecting the intricate structural changes that occur during the binding process, which can lead to biased representations. To address this gap, we propose a novel framework titled "Deep-ComAIR", which effectively models the binding process by focusing on the complex structure of TCR-pMHC rather than individual components. This model enhances prediction accuracy by integrating features from three modalities: sequence, structural, and gene. Our approach achieves state-of-the-art results evidenced by an area under the receiver operating characteristic curve (AUROC) of 0.983 in binding reactivity prediction and a Pearson correlation coefficient of 0.827 in binding affinity prediction. These results highlight the framework's potential to deepen our understanding of TCR-pMHC interactions at the structural level and facilitate advancements in immunotherapy and vaccine design.

## 1 Introduction

T cells play a pivotal role in the immune response. By surveying the presence of foreign peptides presented by major histocompatibility complex (MHC) molecules on the surface of cells, T cells with specific receptors can evaluate the health status of these cells. This interaction, called TCR-pMHC (peptide-MHC) binding, serves as a critical switch for overall immune activation, facilitating the induced activation and proliferation of antigen-specific T cells. Consequently, a deeper exploration of the binding process and the underlying mechanisms is considered essential for advancing personalized immunotherapy.

In most instances, the $\alpha\beta$ T cells, characterized by $\alpha\beta$ T cell receptors (TCRs), participate in the binding process. The TCR comprises an $\alpha$ chain and a $\beta$ chain, which are linked by disulfide bonds and recognize pMHC as a heterodimer. These chains can be categorized into several regions, e.g., constant (C) regions and variable (V) regions. The V regions of the $\alpha$ and $\beta$ chains collectively form the antigen-binding site of the TCR. Furthermore, the V region is subdivided into complementarity-determining regions (CDRs), specifically CDR1, CDR2, and CDR3. Upon recognition of the ligand, the less diverse CDR1 and CDR2 engage with the $\alpha$ helices (helix 1 and helix 2) on either side of the pMHC, while CDR3 interacts with the central peptide. CDR3 is a crucial sequence that significantly influences the specificity of TCR recognition of antigens, with its high diversity arising from the genomic recombination of the variable, diversity, and joining (VDJ) genes. In figure 5, we select several examples to demonstrate TCR-pMHC binding directly.

Several studies have been proposed to predict TCR-pMHC interactions. The task includes two main sub-tasks: binding affinity prediction and binding reactivity prediction. Reactivity prediction refers to whether TCRs bind to the peptide-MHCs. GLIPH and TCRdist utilize statistical methods for the predictions, and RACER uses a pairwise energy model. With the development of artificial intelligence, many studies focus on developing effective deep learning methods and completing the task by solving a classification/regression problem. DeepTCR, TCRAI, NetTCR, TcellMatch, DeepAIR, and other models successfully modeled the binding process and achieved state-of-the-

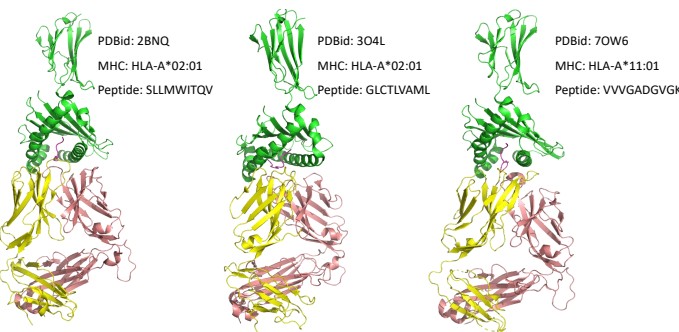

Figure 1: **Binding Instances.** The figure illustrates several examples of binding interactions, each comprising a T-cell receptor (TCR) and a peptide-major histocompatibility complex (pMHC). In these representations, the MHC is depicted in green, while the peptide is shown in magenta. The TCR comprises two chains: the alpha chain is highlighted in yellow, and the beta chain is depicted in pink.

art(SOTA) results. Affinity prediction focuses on the binding strength and is usually considered a further step beyond reactivity prediction.

Early machine learning and deep learning approaches primarily focused on features derived from sequence data. Some of them rely on large models pretrained on large scales of datasets and generate high-level representations for the sequences. However, the structural configuration of the TCR-pMHC complex plays a fundamental role in antigen recognition and binding. Some models have noticed the importance of structure, and the success of protein structure predictors such as AlphaFold2 and ESM-Fold has further motivated the incorporation of structural information to enhance prediction accuracy. Despite achieving state-of-the-art results, these methods typically consider the structures of TCR and pMHC individually, overlooking the structural changes that occur during the binding process. Structural modeling shows that certain amino acids shift positions upon binding, leading to changes in the monomeric structures of both TCR and pMHC, which are critical for determining whether binding will occur.

In response to these challenges, we propose a multi-modal fusion model that integrates pretrained large models with deep learning techniques. Our model encodes sequence and structural information from the TCR-pMHC complex and relevant gene features to generate three distinct feature sets. These features are then fused through a series of forward layers incorporating a residual-like, multi-feature-aware structure. Finally, a classification head is employed to predict binding reactivity, and a regression head for binding affinity.

Our contribution can be illustrated as follows:

- We propose a novel framework for TCR-pMHC binding prediction named Deep-COM AIR. It leverages comprehensive information from three modalities, offering a more holistic understanding of the binding process.
- We account for the subtle structural changes that occur during the binding process and encode structural data more unbiasedly.
- We introduce a residual-like, multi-feature-aware architecture that effectively integrates features from different layers, thereby enhancing the expressive power of the model.

## 2 RELATED WORK

### 2.1 MODELS FOR BINDING PREDICTION

A lot of models have been developed for binding prediction. GLIPH (Glanville et al., 2017) and TCRdist (Dash et al., 2017) used traditional statistical methods for the prediction, while RACER (Lin et al., 2021) utilized supervised machine learning model and introduced a pairwise energy model to solve the problem.

With the advancement of deep learning, deep neural networks have proven effective in binding prediction tasks. DeepTCR (Sidhom et al., 2021) utilized convolutional and embedding layers for T-cell receptor featurization, followed by a variational autoencoder (VAE) to fuse sequence and gene features, modeling the latent space with several Gaussian distributions. TCRAI (Zhang et al., 2021) introduced ICON to reliably identify TCR–peptide–major histocompatibility complex interactions. It employed a network with a more flexible architecture and used batch normalization, to create fingerprints for a given TCR for prediction purposes. SoNNia (Isacchini et al., 2021) leveraged biophysical models of receptor generation combined with machine learning models of selection to identify specific sequence features characteristic of functional lymphocyte repertoires and subrepertoires. ERGO (Springer et al., 2020) encoded the TCR and peptide using long short-term memory (LSTM) and multilayer perceptron (MLP) encoders, training a MLP classifier for binding prediction. NetTCR (Montemurro et al., 2021) takes CDR3 regions and peptide as input only, using a one-dimention convolution layer to encode the embeddings produced by BLOSUM. Some other work also focused on binding prediction and provided complete V(D)J gene sequencing data. (Fischer et al., 2020; Lu et al., 2021; Zaslavsky et al., 2022; Widrich et al., 2020)

With the development of pretrained models, features are extracted by models like ProtBert, ESM, and AlphaFold. SC-AIR-BERT (Zhao et al., 2023b) learns the "language" of antigen receptor (AIR) sequences through self-supervised pre-training on a large cohort of paired AIR chains from multiple single-cell resources. The model is then fine-tuned with a multilayer perceptron head for binding specificity prediction, employing a k-mer strategy to enhance sequence representation learning. DeepAIR (Zhao et al., 2023a) takes a further step by introducing structural information. The model fuses features from the structure and sequence of the TCR-peptide and the relevant gene. However, though CDR3 loops may be primarily responsible for antigen recognition, residues from CDR1, CDR2 and even the framework region of both $\alpha$-chains and $\beta$-chains may be involved . DeepAIR failed to capture the information provided by the peptide-MHC complex and ignored small structural changes of the TCR-pMHC complex, which may lead to bias in binding prediction. Also, there exists some other problems, such as the absence of experimental negatives. (Hudson et al., 2023)

## 2.2 Pretrained Models for Protein Feature Representation

Pretrained models refer to those models trained on large scale of dateset. ProtBert (Brandes et al., 2022) is based on Bert structure. It considered protein sequences as a kind of language and was pretrained on a large corpus of protein sequences in a self-supervised fashion using a masked language modeling(MLM) objective. Evolutionary Scale Modeling (ESM) (Lin et al., 2023) is a series of transformer-based models designed for protein sequence representation and structure prediction. Model of 15 billion parameters is trained over sequences in the UniRef (32) protein sequence database, also with MLM objective. During training sequences are sampled with even weighting across around 43 million UniRef50 training clusters from around 138 million UniRef90 sequences so that over the course of training the model sees around 65 million unique sequences. Foldseek (Van Kempen et al., 2024) developed a novel type of structural alphabet and enables effective structure comparison and search.The 20 states of the 3D-interactions (3Di) alphabet describe for each residue i the geometric conformation with its spatially closest residue j.By representing the amino acid backbone of proteins as sequences over a structural alphabet, it enables faster structures comparison by reducing it into faster sequence alignments.

## 3 Methods

In this section, we will introduce our methods for binding prediction of TCR-pMHC complex. We begin by introducing the problem setup in Section 3.1. Next, we elucidate multi-modal data encoder in Section 3.2. Finally, we describe the data fusion procedure, with a specific multimodal feature fusion module, in Section 3.3. The overall architecture of the proposed methods is illustrated in Figure 2.

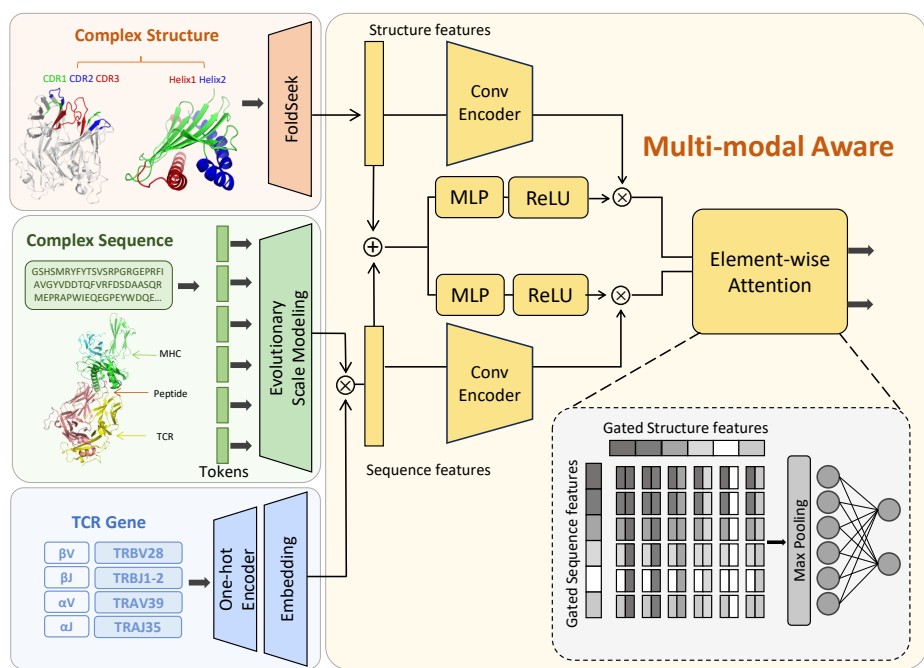

Figure 2: **Overall architecture of the proposed DEEP-COMAIR.** This framework delineates how we encode the samples from three modals, namely sequence, structure and gene, and fuse them through a residual-like multi-feature fusion layer and a multimodal-attention layer.

## 3.1 PROBLEM SETUP

Our purpose is to predict the binding reactivity and affinity between TCR and pMHC. Let $T = \{t_1, t_2, \ldots, t_m\}$ represent m TCRs, each of which contains alpha (light) sequence and beta (heavy) sequence with CDR regions separately, and $M = \{m_1, m_2, \cdots, m_n\}$ represents n peptide-MHC.

The overall goal is set into two steps. In the first step, the reactivity samples $S_r = \{s_{r1}, s_{r2}, \cdots, s_{rn}\}$ are as follows:

$$s_{\text{positive, k}} = (t_{p1}, m_{p2}; 1)$$
$$s_{\text{negative k}} = (t_{n1}, m_{n2}; 0) \tag{1}$$

with 0 and 1 representing not binding/binding. A classification model is utilized to predict the binding reactivity of TCR-pMHC and divides them into two categories. In the second step, we use another regression model to predict the binding affinity (the value of UMI), and the affinity samples are $S_a = \{s_{a1}, s_{a2} \cdots, s_{an}\}$, where

$$s_{\text{umi, k}} = (t_i, m_j; \text{umi value}) \tag{2}$$

## 3.2 MULTI-MODAL DATA ENCODER

We employ a tri-modal encoding framework for the TCR-pMHC complex, incorporating three distinct feature encoders: the V(D)J gene encoder, the sequence encoder, and the structure encoder.

The sequence encoder derives high-level representations of the sequence information for the two TCR chains (alpha and beta sequences), peptides, and the helix1/helix2 sequences of MHC. This is achieved through ESM-2 (Evolutionary Scale Modeling 2), a pretrained multilayer transformer encoder. ESM-2(650M) is composed of 33 transformer layers with 20 attention heads and an embedding dimension of 1280. Trained on large protein sequence datasets, including sampled UniRef50/90

(65 million proteins), it utilizes a masked language modeling (MLM) objective. During MLM self-supervised training, the model is tasked with predicting the identity of randomly masked amino acids in a sequence by observing the context in the rest of the sequence. This allows the model to capture intricate dependencies between amino acids, producing an informative feature representation of the entire protein sequence. In the sequence encoder of Deep-ComAIR, different from sequence-based models such as DeepTCR and multi-modal model DeepAIR, sequences of varying lengths are encoded and passed through a max-pooling layer in the feature dimension.

For structural encoding, we leverage Foldseek to produce 3D immersive (3di) embeddings. Foldseek generates discrete states for each residue based on their local 3D spatial arrangement, specifically calculating a 10-dimensional descriptor that encapsulates the conformation of the residue's local backbone and its nearest neighbors. The embeddings are generated through the encoder component of a vector quantized variational autoencoder (VQ-VAE), offering a more detailed representation of tertiary contacts between residues. This sensitivity to subtle alterations in residue conformation, frequently observed during binding, plays a crucial role in understanding the binding process. The structural encoder processes inputs including the CDR regions of the alpha/beta sequences, peptides, and helix1/helix2 segments. Gene modalities, including alpha-v/j genes and beta-v/j genes, are encoded as one-hot tensors, with the position of "1" in the tensor array signifying variations in the alpha-V/alpha-J and beta-V/beta-J gene segments.

### 3.3 MULTI-MODAL FUSION

Following the multimodal feature encoder, the multimodal feature fusion module integrates the features extracted from each channel using a gating-based attention mechanism, along with tensor fusion, to generate a comprehensive representation of the TCR and pMHC complex. Subsequently, task-specific prediction layers map this integrated receptor representation to the final predicted outcomes.

More specifically, let the features obtained from the gene, sequence, and structure encoders be denoted as $f_g$, $f_{seq}$, and $f_{stru}$, respectively. These features are first processed to obtain high-level representations for each modality:

$$
\begin{aligned}
h_{gene} &= \text{Maxpool}(\text{Emb}(f_{gene})) \\
h_{seq} &= \text{Maxpool}(\text{Norm}(\text{Conv}(f'_{seq}))) \\
h_{stru} &= \text{Maxpool}(\text{Norm}(\text{Conv}(f_{struc})))
\end{aligned}
\tag{3}
$$

in which $f'_{seq}$ is obtained by concating five sequence representations (generated by ESM-2) at feature dimensions. We then concatenate $h_g$ and $h_{seq}$ to obtain a synthesized feature, denoted as $h_{bio}$, which will be utilized in the subsequent steps.

To align the features and mitigate the impact of noisy components in $h_{bio}$ and $h_{stru}$ during multi-modal feature fusion, we first apply a residual-like, multi-feature-aware layer to both $h_{seq}$ and $h_{stru}$ individually. These features are then fused using an element-wise attention mechanism. The details are as follows:

$$
h'_{\text{bio}} = a_{\text{bio}} * \hat{h}_{\text{bio}}
\tag{4}
$$

where:

$$
a_{\text{bio}} = \sigma(W_{\text{bs}}^{\text{bio}} \cdot [h_{\text{bio}}, h_{\text{stru}}])
\tag{5}
$$

and

$$
\hat{h}_{\text{bio}} = \text{ReLU}(W_{\text{bio}} \cdot h_{\text{bio}})
\tag{6}
$$

Similarly, we can obtain $h_{\text{stru}}$:

$$
h'_{\text{stru}} = a_{\text{stru}} * \hat{h}_{\text{stru}}
\tag{7}
$$

where:

$$
a_{\text{stru}} = \sigma(W_{bs}^{\text{stru}} \cdot [h_{\text{stru}}, h_{\text{stru}}])
\tag{8}
$$

and

$$
\hat{h}_{\text{stru}} = \text{ReLU}(W_{\text{stru}} \cdot h_{\text{stru}})
\tag{9}
$$

where $W_{\text{bs}}^{\text{bio}}, W_{\text{bio}}, W_{bs}^{\text{stru}}, W_{\text{stru}}$ are trainable weight matrix parameters from four multilayer perceptrons (MLPs), and $\sigma$ represents the sigmoid activation function. This is called a residual-like multi-feature aware layer, or a gated-based attention mechanism to adjust the expressiveness of them by

attention score vector $a_{\text{bio}}$ and $a_{\text{stru}}$. By incorporating structure-sequence-gene fusion information into the individual encoding processes of $h_{\text{bio}}$ and $h_{\text{stru}}$, we effectively combine and align the features, obtaining a high-level embedding of the modalities. Finally, $h'\text{bio}$ and $h'\text{stru}$ are passed into the element-wise attention module:

$$h_{\text{final}} = \text{MM}([h'_{\text{bio}}, 1], [h'_{\text{stru}}, -1]) \tag{10}$$

where MM refers to matrix multiplication. The concatenated values 1 and -1 are used to distinguish between the different modalities. This element-wise attention module requires each feature to be aware of all features in the other modality, thereby enabling the generation of a comprehensive representation of the TCR-pMHC complex. $h_{\text{final}}$ is used for binding reactivity and affinity prediction through a classification head or a regression head.

## 4 EXPERIMENTS

### 4.1 DATASET AND NEGATIVE SAMPLES CONSTRUCTION

We utilized several sources of data to train our model, including single-cell TCR data captured by pMHC multimers from the 10x Genomics website and VDJdb, a curated database of T-cell receptor (TCR) sequences with known antigen specificities collected from published results of TCR specificity assays and personal communications and submissions.

For the binding reactivity task, we have 8613 pairs of TCR-pMHC data for high-confidence data and 25717 pairs for low-confidence data. Every data is featured by specific amino acid sequences, the TCR gene, and other relative features. Difference between high-confidence and low-confidence data exists in the feature "score". The higher the score is, the more confidence we have in the antigen specificity annotation of a given TCR. A zero score indicates insufficient details regarding the method used to draw a conclusion. The high-confidence dataset excludes data with a score of 0, while the low-confidence dataset includes them.

Since the TCR–pMHC dataset contains only positive samples, in order to train a generalized and robust supervised model, the negative samples are required and should be generated via a biologically and computationally plausible manner to serve as an unbiased estimate of the actual distribution of non-binding pairs. Here, we follow the most common method of shuffling the known positive pairs. To be precise, each CDR3 sample in the positive dataset was combined with a peptide-MHC complex from another positive sample to make negative samples. Shuffling known positive pairs makes the assumption that TCRs can only bind to their seen epitope partner, and that any other combination would not lead to successful binding events, which is plausible considering the enormous diversity of potential TCR and peptide-MHC sequences.



Figure 3: **Negative samples** are constructed by shuffling the positive pairs.

We constructed two datasets of different properties of negative samples, in which negative samples are equal and nine times the positive ones. Test dataset is ensured that the number of positive samples equals to the negative ones.

For the binding affinity task, we constructed 5592 pairs of TCR-pMHC data with umi values, and all the data can be found in the high-confidence reactivity dataset. The number is smaller because some of the data can not be successfully modeled.

## 4.2 TRAINING DETAILS

**Setting.** For the optimization of model parameters, we employed the AdamW optimizer with a learning rate of 1e-4, weight decay of 0.001, and the default parameters of $\beta_1 = 0.9$, $\beta_2 = 0.999$, and $\epsilon = 1e - 8$. All parameters were initialized by He initialize (He et al., 2015) for better optimization, together with the using of ReLU activation function. The batchsize was set to 32, with the number of training epoch set to 50. All experiments were completed on single 4090 GPU.

**Loss Function.** For the task of predicting binding reactivity, which is formulated as a binary classification problem, we employed a combination of cross-entropy (CE) loss and mean squared error (MSE) loss, with an equal weighting ratio of 1:1. The cross-entropy loss is utilized to effectively capture the divergence between the predicted probabilities and the ground truth labels, while the MSE loss serves to minimize the overall error magnitude in the logits, thereby enhancing prediction robustness. The total loss for this task is defined as:

$$L_{\text{reactivity}} = L_{\text{CE}} + L_{\text{MSE}} \tag{11}$$

The second task of binding affinity prediction is a regression problem. For this task, we adopted a hybrid loss function comprising cross-entropy loss, MSE loss, and a regularization term. Given the large value of the regression target, a logarithmic transformation was applied to both the predictions and the true labels prior to calculating the loss. MSE was then computed to quantify the discrepancy between the predicted and true values. In addition, the log-transformed outputs were mapped into eight intervals, treating the problem as a classification task, where cross-entropy loss was applied accordingly. To further regulate model complexity and prevent overfitting, a regularization term was included, based on the L2 norm of the classification head parameters. The final loss function balances the contributions of MSE, cross-entropy, and the regularization term in a ratio of 1:1:0.01. This approach ensures that both regression precision and classification alignment are preserved. The use of cross-entropy loss aids in achieving better convergence and mitigates the risk of mode collapse, while the regularization term helps manage overfitting by constraining model complexity. The total loss for this task is defined as:

$$L_{\text{affinity}} = L_{\text{CE}} + L_{\text{MSE}} + 0.01 * L_{\text{Regu}} \tag{12}$$

**Evalution** We evaluated the model's performance using accuracy metrics and the correlation between the predicted and true values. For the binding reactivity task, we computed the area under the receiver operating characteristic curve (AUROC) on the test dataset. To ensure the validity of the evaluation, the test dataset was carefully balanced with an equal number of positive and negative samples, and a rigorous double-checking process was employed to prevent any overlap between the training and test datasets. For the binding affinity task, the dataset was divided, with 80% of the data (4961 samples) used for training and the remaining 20% (551 samples) reserved for testing. Pearson correlation, Spearman correlation and $R^2$ square were calculated to evaluate the performance.

**Baseline** For binding reactivity prediction, we conducted a comprehensive comparison of Deep-ComAIR with several competitive baseline models to assess its performance. Deep-ComAIR leverages features from three modalities—sequence, structure, and gene information—while employing a similar architecture. Baseline models also included TCRAI, DeepTCR, and soNNia, each of which represents a deep-learning method in TCR-pMHC binding prediction. In addition to comparing the full multimodal Deep-ComAIR, we also evaluated its modality-specific variants: DeepAIR-seq, which utilizes only sequence-based features, and DeepAIR-struc, which focuses solely on structural features. These comparisons allowed us to assess the impact of each individual modality and the effectiveness of multimodal fusion in boosting prediction accuracy. For the binding affinity prediction task, we selected DeepAIR and DeepTCR as baseline models. Since this is a regression task, the primary evaluation metric we focused on was the Pearson correlation coefficient.

## 5 RESULTS

### 5.1 RESULTS OF BINDING REACTIVITY PREDICTION

We test the area under the receiver operating characteristic curve (AUROC) on the validated high-confidence dataset and low-confidence dataset.The results are listed in table 1.Our model get State-Of-The-Art(SOTA) performance on all systems categorized by the type of peptides and also on the average level.

Table 1: **Binding Reactivity Prediction AUROC Results on High-confidence and Low-confidence datasets**

| Antigen | | AUROC | | | | | |
|---|---|---|---|---|---|---|---|
| | | DeepCOM-AIR | | DeepAIR | TCRAI | DeepTCR | soNNia |
| Peptide | Source | high-conf | low-conf | | | | |
| AVFDRKSDAK | EBNA4 (EBV) | 0.911 | **0.967** | 0.881 | 0.647 | 0.674 | 0.693 |
| GILGFVFTL | M1 (flu) | 0.962 | **0.993** | 0.955 | 0.938 | 0.929 | 0.840 |
| IVTDFSVIK | EBNA3B (EBV) | 0.970 | **0.979** | 0.922 | 0.835 | 0.847 | 0.674 |
| RAKFKQLL | BZLF1 (EBV) | 0.944 | **0.979** | 0.934 | 0.933 | 0.911 | 0.860 |
| GLCTLVAML | BMLF1 (EBV) | 0.903 | **0.996** | 0.972 | 0.908 | 0.840 | 0.916 |
| ELAGIGILTV | MART-1 (melanoma) | 0.928 | **0.995** | 0.983 | 0.988 | 0.986 | 0.844 |
| KLGGALQAK | IE1 (CMV) | 0.926 | **0.985** | 0.870 | 0.854 | 0.851 | 0.748 |
| **average** | | **0.935** | **0.983** | 0.904 | 0.845 | 0.844 | 0.782 |

### 5.2 COMPARISON OF DIFFERENT STRUCTURE ENCODERS

We selected Foldseek as the structural encoder due to its superior capability in capturing subtle structural variations. To validate its advantages, we conducted experiments on high-confidence dataset, comparing the performance of Foldseek and ESM-Fold as structural encoders, while keeping all other components of the model unchanged. Consistent with the results obtained in the binding reactivity prediction task, we calculated the AUROC for each peptide as well as the overall system. The results, as presented in Table 2, demonstrate that employing Foldseek as the structural encoder leads to improved performance, attributable to its heightened sensitivity to small structural changes.

Table 2: **Binding Reactivity Prediction AUROC Results on High-confidence datasets using different structure encoders**

| Antigen | | AUROC | |
|---|---|---|---|
| | | DeepCOM-AIR | |
| Peptide | Source | foldseek | ESM-fold |
| AVFDRKSDAK | EBNA4 (EBV) | **0.911** | 0.804 |
| GILGFVFTL | M1 (flu) | **0.962** | 0.856 |
| IVTDFSVIK | EBNA3B (EBV) | **0.970** | 0.650 |
| RAKFKQLL | BZLF1 (EBV) | **0.944** | 0.756 |
| GLCTLVAML | BMLF1 (EBV) | **0.903** | 0.761 |
| ELAGIGILTV | MART-1 (melanoma) | **0.928** | 0.809 |
| KLGGALQAK | IE1 (CMV) | **0.926** | 0.681 |
| **average** | | **0.935** | 0.748 |

### 5.3 COMPARISON OF DIFFERENT MODAL

We compared the performance of models utilizing the fusion of three modalities (sequence, structure, and gene features) with those relying solely on sequence or structure features. These experiments were conducted on low-confidence datasets to better assess the robustness of each model configuration. The results highlight the importance of multimodal fusion, as it significantly enhances prediction accuracy.

As shown in Table 3, most methods demonstrated superior performance when predicting TCRs that specifically bind to ELAGIGILTV (a peptide from the MART-1 protein, associated with melanoma),

Table 3: **Binding Reactivity Prediction AUROC Results on High-confidence dataset** Comparison of models utilizing the fusion of three modalities and relying on sequence or structure only.

| Antigen | | AUROC | | | | | |
| --- | --- | --- | --- | --- | --- | --- | --- |
| | | DeepCOM-AIR | | | DeepAIR | | |
| **Peptide** | **Source** | MM | struc | seq | MM | struc | seq |
| AVFDRKSDAK | EBNA4 (EBV) | **0.967** | 0.946 | 0.544 | 0.881 | 0.738 | 0.598 |
| GILGFVFTL | M1 (flu) | **0.993** | 0.976 | 0.819 | 0.955 | 0.940 | 0.933 |
| IVTDFSVIK | EBNA3B (EBV) | **0.979** | 0.977 | 0.684 | 0.922 | 0.885 | 0.807 |
| RAKFKQLL | BZLF1 (EBV) | **0.979** | 0.957 | 0.809 | 0.934 | 0.907 | 0.879 |
| GLCTLVAML | BMLF1 (EBV) | **0.996** | 0.995 | 0.909 | 0.972 | 0.876 | 0.912 |
| ELAGIGILTV | MART-1 (melanoma) | **0.995** | 0.976 | 0.944 | 0.983 | 0.938 | 0.960 |
| KLGGALQAK | IE1 (CMV) | **0.985** | 0.970 | 0.545 | 0.870 | 0.858 | 0.768 |
| **average** | | **0.983** | 0.964 | 0.640 | 0.904 | 0.867 | 0.827 |

whereas their performance was notably lower for TCRs binding to AVFDRKSDAK (from the EBNA4 protein of EBV) and KLGGALQAK (from the IE1 protein of CMV). These findings suggest that the TCRs recognizing AVFDRKSDAK and KLGGALQAK exhibit greater diversity compared to those recognizing ELAGIGILTV, making the former more challenging to predict.

Importantly, the introduction of structural information markedly improved the performance for AVF-DRKSDAK and KLGGALQAK, underscoring the critical role that structural features play in accurately predicting binding reactivity. This demonstrates that incorporating structural data is essential for capturing the complexities of TCR-peptide interactions, especially in cases where sequence features alone may not provide sufficient resolution for accurate prediction.

## 5.4 RESULTS OF BINDING AFFINITY PREDICTION

We calculate the Pearson correlation coefficient and Spearman's rank correlation coefficient between the prediction umi value and the ground truth value. By comparison of the existing models, our method achieves the best performance, with a Pearson correlation coefficient of **0.82**, Spearman's rank correlation coefficient 0f **0.70** and $R^2$ score of **0.60**.

Figure 4: **Binding Affinity Prediction Results**

Figure 5: **Regression curve.** Visualization of predicted value and true values for affinity prediction.

| Model | Pearson's Correlation |
| --- | --- |
| **DeepCOM-AIR** | **0.829** |
| **DeepAIR** | 0.813 |
| **TCRAI** | / |
| **DeepTCR** | 0.754 |
| **soNNia** | / |

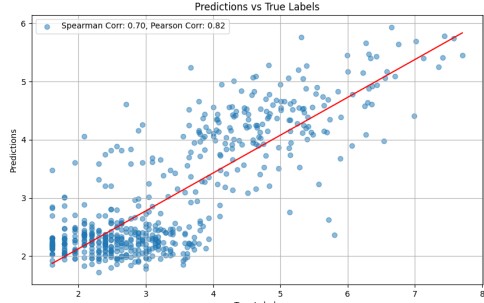

## 6 CONCLUSION

In this study, we addressed the challenge of predicting binding reactivity and affinity within the context of TCR-pMHC interactions. We examined the potential biases that may arise during the encoding process, which can contribute to erroneous predictions. To mitigate these challenges, we introduced a novel framework, Deep-ComAIR, which leverages comprehensive information from the TCR-pMHC complex structure. The model was evaluated on diverse multi-source datasets, consistently demonstrating superior performance across a range of predictive tasks. Our results underscore the utility of Deep-ComAIR in advancing adaptive immune receptor analysis, offering

a robust tool for elucidating the mechanisms underlying immune recognition. Furthermore, the framework holds significant promise for facilitating research in personalized immunotherapy by enhancing the accuracy of affinity predictions in TCR-pMHC binding.

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
