# OpenReview forum: "Deep-ComAIR: A Framework for Predicting TCR-pMHC Binding through Complex Structural Analysis"
_ICLR.cc/2025/Conference — Submitted to ICLR 2025_

### Official Review · Reviewer_UTo2 · 2024-11-01

**Soundness:** 1
**Presentation:** 2
**Contribution:** 2
**Rating:** 3
**Confidence:** 4

**Summary:**

This paper presents a model for prediction of TCR-pMHC binding reactivity prediction and binding affinity prediction. It uses three modalities an inputs: sequence, structure, and gene. It utilises the structure of the entire TCR-pMHC complex.

**Strengths:**

This paper provides a good review of existing TCR-pMC binding models, and the classification of such models as (a) traditional statistical methods, (b) deep neural networks trained from scratch, or (c) based on large pretrained models is helpful. The idea of using the structure of the entire TCR-pMHC complex is a good one, and FoldSeek is an appropriate method of featurising structures, which is shown to outperform ESM-fold. Removing either sequence or structure information from the model in this paper damages performance, which suggests that both sequence and structure are being used encoded in a relevant way.

**Weaknesses:**

The paper claims to utilise "several sources of data to train our model", but is scant on further details, other than that it includes data from 10x Genomics and VDJdb. However, these data do not appear to include structure data. No details are provided on how the structure data for training the model is obtained. Since this paper attributes most of the performance to structure data (moreover full complex structure data), this is a significant oversight. Similarly, the model is trained to predict binding affinity labels, but it is not clear how these binding affinity data are obtained, or how confidence scores are given for structural complexes. It is unclear whether the test set is a high-confidence or low-confidence dataset.

Some details are lacking to reproduce the model, such as how "sequences of varying lengths" are generated. It is not clear whether $L_\text{reactivity}$ and $L_\text{affinity}$ are combined to train the model, or whether there are two different models.

The experiments to assess performance of Deep-ComAIR are limited. One of the major challenges in TCR-pMHC binding prediction is generalisation to pMHCs with little or no binding data, but the model is only assessed on a handful of peptides for which there is a wealth of data. Further ablation studies to the model would also be useful, to demonstrate the effect of using pretrained sequence representations and V/J gene labels. No ablation study is performed on the binding affinity prediction task.

The paper contains several typographical and grammatical errors.

**Questions:**

1. How was the structure data for training collected? How was the binding affinity data collected?

2. Which test set is used for the experiments - high or low confidence?

3. How does the model perform for pMHCs in the low data regime? How does this compare with the other models in the results section?

4. How important are pretrained sequence representations for model performance? What about gene labels?

5. Are the binding prediction and binding affinity models trained separately? Does the model benefit from transfer learning between the two tasks?

---

### Official Review · Reviewer_MMhD · 2024-11-03

**Soundness:** 2
**Presentation:** 2
**Contribution:** 1
**Rating:** 3
**Confidence:** 4

**Summary:**

The topic is essential since the binding process between TCR and pMHC is a fundamental mechanism in adaptive immunity. This paper follows DeepAIR and slightly changed the framework to form the new method Deep-COM AIR. Though the authors show that the developed algorithm outperforms other state-of-the-art prediction tools by accounting for the subtle structural changes that occur during the binding process and encoding structural data more unbiasedly, it is not clear where the improvements are from. The paper provides a comprehensive overview of the problem, and related work.

**Strengths:**

1. This paper proposes a framework for predicting TCR-pMHC binding and outperforms the comparison methods on the test set.
2. This paper leverages comprehensive information from three modalities and fuses them through a series of forward layers incorporating a residual-like, multi-feature-aware structure.

**Weaknesses:**

1. The manuscript doesn't show enough novelty. They claimed to  include the intricate structural changes, and used the gene encoder. Though I don't think the gene encoder could in fact solve the problem (as also indicated in DeepAIR), the authors didn't address the improvement by the inclusion.  The authors didn't introduce how to generate the complex structures (I'm assuming that they followed DeepAIR to generate structures through AF2), then what's the difference between DeepAIR.
2. The comparison method DeepAIR also utilizes the same three modalities as mentioned in the paper. Was a comparison conducted under the same context as DeepAIR?
3. In Method section 3.3, the description of the Element-wise Attention module is vague, reducing it to basic matrix multiplication without clarifying whether it involves more complexity or specific design choices. Is the element-wise attention module just a matrix multiplication, or are there other details that haven't been explained?
4. The model design for the baseline that ablates multimodal information is not detailed enough. How is the model for DeepAIR-seq, which utilizes only sequence-based features, designed?
5. The paper only presents results on the test sets without showing training results or using 5-fold cross-validation.
6. The authors do not provide code for reproduction.
7. The "five sequence representations" in line 249 are unclear.

**Questions:**

The authors need to show their novelty. Currently, the inclusion of the module to add  the intricate structural changes , is not novel and not very useful either (as also indicated in DeepAIR).

---

### Official Review · Reviewer_KVEU · 2024-11-04

**Soundness:** 3
**Presentation:** 2
**Contribution:** 2
**Rating:** 6
**Confidence:** 4

**Summary:**

The paper introduces Deep-ComAIR, a framework designed for predicting TCR-pMHC (T cell receptor - peptide-major histocompatibility complex) binding by focusing on the complex structural interactions within the binding process. Unlike previous models that separately analyse sequence or structural features of TCR and pMHC, Deep-ComAIR integrates sequence, structural, and gene-based features to capture nuanced structural changes (on complex formation) that occur upon binding. This framework advances TCR-pMHC interaction predictions and could have potential applications in immunotherapy and vaccine development.

**Strengths:**

The paper brings a novel multimodal approach to TCR-pMHC binding prediction by incorporating structural changes on complex formation that are often overlooked in previous models.

**Weaknesses:**

- The proposed model is largely an incremental work on DeepAIR model. This model mainly differs in terms of how the structure is encoded. It would be interesting to see how much difference using the structural embeddings of the complex instead of individual structures makes in the model performance. A direct comparison with the DeepAIR model for some examples of TCR/pMHCs pairs can be done to bring home the point about the importance of using the complex-structures. The same architecture (multimodal fusion + gated-attention), but with structural embeddings of individual monomers (instead of the complex) can also be used for comparison.

- The current model is a black-box model which doesn't highlight which part of the sequence/structure are important for the binding. The attention weights can be used to highlight the residues that are important for the binding.

- The structure of the TCR might not always be available. Foldseek relies on AlphaFold 2 structures for the structural embeddings, but they aren't always accurate. The dependence of the model performance on the structural model quality should be investigated. pLDDT scores from AlphaFold models can be used to this analysis. If there are experimental structures available in the PDB, using those structures instead of the predicted model might be more accurate.

- Recent studies on TCR-pMHC prediction models suggest that these models aren't generalizable and have a strong data dependency. Therefore a more comprehensive benchamrking is needed to ascertain the performance of the models. This includes testing on multiple datasets and also looking at the peptide distributions of the training/testing sets. The authors have used the 10x Genomics website-data and VDJdb-database in this study. More testing can be done on datasets from McPAS-TCR, ImmuneCODE, and IEDB

**Questions:**

- It's not clear from the paper that how are the sequence embeddings derived from ESM2. I would assume that authors derived a fixed length embedding of 1280 dimensions per TCR-sequence (averaged over all residues). How were the TCR-sequence embeddings combined with the pMHC-sequence embeddings? Are the embeddings for TCR-sequence and pMHC-sequence derived separately and then combined?

- The information about how training and testing sets were split is missing. In the 'DATASET AND NEGATIVE SAMPLES CONSTRUCTION' section, there needs to be information about the training and testing datasets. For sequence-based tasks, it's also important to look at the homology of the sequences and there isn't a large overlap between the training and testing dataset. The authors can look at tools like CD-HIT to ensure that the sequences in the testing and training-datasets are non-homologous (70 to 80% homology cut-off)

- An additional suggestion to the authors would be to try some additional pre-trained models for encoding the strcuture/sequence. There are newer models such as ESM3, ProstT5, both of which now encode structure and sequence.

---

### Official Review · Reviewer_vA8g · 2024-11-09

**Soundness:** 3
**Presentation:** 3
**Contribution:** 2
**Rating:** 3
**Confidence:** 3

**Summary:**

This paper introduces a novel framework for predicting the binding between T cell receptors (TCRs) and peptide-major histocompatibility complexes (pMHCs), a critical process in adaptive immunity. The Deep-ComAIR framework focuses on the complex structure of the TCR-pMHC interaction rather than just individual components, enhancing prediction accuracy by integrating multimodal features, such as sequence, structural, and gene.

**Strengths:**

Using multimodal features, such as sequence, structural, and gene, helps to improve the performance. Ablation is good.

**Weaknesses:**

The method seems to be trivial for AI conference, and I do not see some innovative algorithm design. The problem definition itself may be questionable: how can we got the complex structure in realistic scenarios? The complex conformation should be unknown and need to be predicted. This paper oversimplifies the problem.

**Questions:**

Q1: How can we got the complex structure in realistic scenarios?

Q2: When mentioning the dataset, why not give them appropriate citations?

Q3: Have you tried to use neural networks to encode the structure coordinates instead of 3Di tokens? What the performance will be?

Q4: How do you use ESMFold to encode structures?

---

### Meta-Review · Area_Chair_KjWm · 2024-12-21

**Metareview:**

The paper contributes a novel representation of T-cell to better reflect the underlying biological processes.
The reviewers were unable to see the strengths in the paper for it be published in ICLR.
The author(s) did not submit any rebuttals.

**Additional Comments On Reviewer Discussion:**

The author(s) did not submit any rebuttal

---

### Decision · Program_Chairs · 2025-01-22

Reject